

# Transcriptome sequencing and comparative analysis of adult ovary and testis identify potential gonadal maintenance-related genes in *Mauremys reevesii* with temperature-dependent sex determination

Lei Xiong[1,2], Jinxiu Dong[1], Hui Jiang[1], Jiawei Zan[1], Jiucui Tong[1,2], Jianjun Liu[1], Meng Wang[1] and Liuwang Nie[1]

[1] Life Science College of Anhui Normal University, Provincial Key Lab of the Conservation and Exploitation Research of Biological Resources in Anhui, Wuhu, Anhui, P.R. China
[2] Biochemistry Department of Wannan Medical College, Provincial Key Laboratory of Biological Macro-molecules Research, Wuhu, Anhui, P.R. China

Corresponding author
Liuwang Nie,
lwnie@mail.ahnu.edu.cn

## ABSTRACT

*Mauremys reevesii* is a classical organism with temperature-dependent sex determination (TSD). Gonad development in early life has recently received considerable attention but gonadal maintenance after sex differentiation in turtles with TSD remains a mystery. In this study, we sequenced the transcriptomes for the adult testis and ovary using RNA-seq, and 36,221 transcripts were identified. In total, 1,594 differentially expressed genes (DEGs) were identified where 756 DEGs were upregulated in the testis and 838 DEGs were upregulated in the ovary. Gene Ontology and Kyoto Encyclopedia of Genes and Genomes pathway analysis suggested that the TGF-beta signaling pathway and Hedgehog signaling pathway have important roles in testis maintenance and spermatogenesis, whereas the Hippo signaling pathway and Wnt signaling pathway are likely to participate in ovary maintenance. We determined the existence of antagonistic networks containing significant specific-expressed genes and pathways related to gonadal maintenance and gametogenesis in the adult gonads of *M. reevesii*. The candidate gene Fibronectin type 3 and ankyrin repeat domains 1 (*FANK1*) might be involved with the regulation of testis spermatogenesis.

## INTRODUCTION

*Mauremys reevesii*, also called Reeves' pond turtle, is one of the most common and widespread semiaquatic turtles in East Asia (*Yin et al., 2016*). This species is a classical organism with temperature-dependent sex determination (TSD). Previous studies developed a gene expression model for the embryonic gonads in reptiles with TSD (*Yatsu et al., 2016*; *Czerwinski et al., 2016*), and many of the sex-related genes have also

been identified, for example, *DMRT1* in *Pelodiscus sinensis* and *Trachemys scripta* (*Sun et al., 2017*; *Ge et al., 2017*), and *CYP19A1* in *T. scripta* (*Matsumoto et al., 2013*). In species with significant sex chromosomal differentiation, gonadal maintenance after sex differentiation depends on the expression of XX/XY or ZZ/ZW genes. However, species with TSD have almost identical genetic materials in the males and females. Thus, the mechanisms responsible for functional maintenance and gametogenesis during the sexual maturity period remain a mystery. Transcriptome sequencing provides a general representation of most of the transcripts that are expressed in specific cells or organs under particular conditions and stages (*Liu et al., 2018*), and it is considered the best option for identifying candidate genes in organisms that lack a reference genome (*Yin et al., 2016*).

In this study, we performed a comparative transcriptome analysis of the adult gonads (testis and ovary) in *M. reevesii* in order to identify molecular signaling cascades and gene expression networks. We also annotated all of the sequenced transcripts and enriched differentially expressed genes (DEGs), and pathways related to gonadal maintenance in these two sexual organs. These results provide crucial genomic information to facilitate further research into the regulatory mechanisms related to gonadal maintenance and gametogenesis during sexual maturity in turtle species with TSD.

## MATERIALS AND METHODS

### Ethical approval

Procedures involving animals and their care were approved by the Animal Care and Use Committee of Anhui Normal University (#20170612).

### Sample collection and RNA extraction

Six adult turtles (3♂, 3♀) were cultivated under the same rearing conditions (i.e., maintenance in a room at temperatures of 10–30 °C) in the Provincial Key Laboratory for the Conservation and Exploitation Research of Biological Resources in Wuhu (31°33′N, 118°370′E, southeast of China) in 2017. The turtles were aged more than 6 years old in the sexual maturity stage, and the sexes of the samples were determined by dissection. Gonad samples (testicles and ovaries) were collected separately and dissected (Fig. 1). The gonad samples were flash frozen in liquid nitrogen and stored at −80 °C until RNA was extracted (*Yin et al., 2016*).

RNA was extracted from each sample with TRIzol reagent (Invitrogen, Carlsbad, CA, USA) according to the manufacturer's instructions. To effectively remove the genomic DNA, we added RNase-free DNase I (Takara, Dalian, China) to the reaction mixture for at least 10 min. The extracted RNA was quantified with the Nanodrop system (Thermo, Wilmington, DE, USA) and the fragment size distribution were checked by 1.5% agarose gel electrophoresis. RNA from each sample was used for cDNA synthesis and sequencing.

### cDNA library construction and sequencing

We used 15 µg of total RNA for cDNA library construction using a TruSeq®RNA LT Sample Prep Kit v2 (Illumina, San Diego, CA, USA) according to the manufacturer's protocol.

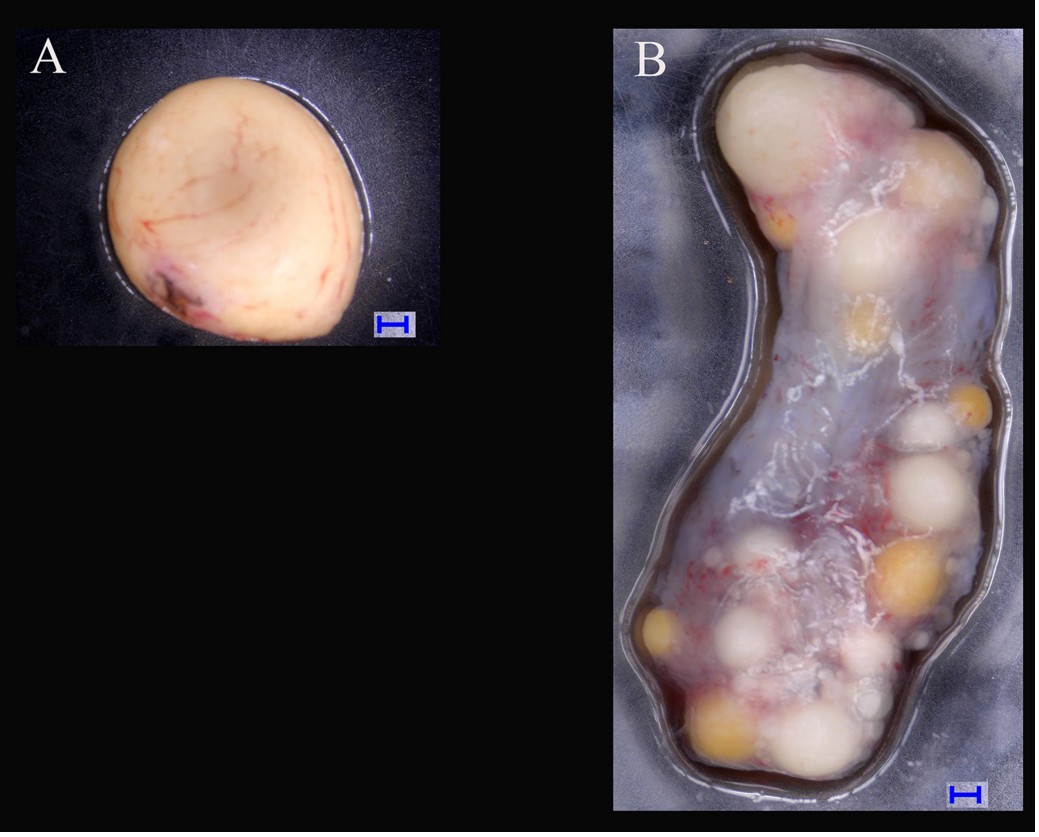

**Figure 1 Tissues of ovary and testis in *Mauremys reevesii* used in deep sequencing.** (A) Testis; (B) Ovary. Blue bar = 1,000 μm; Digital microscopic system VHX-5000.

Six cDNA libraries (3♂, 3♀) were constructed by performing end-repair, 3′-end adenylation, and adapter ligation and enrichment. Sequencing was performed using the Illumina Hiseq 2500 platform by Genergy Bio-technology Co. Ltd (Shanghai, China). All sequence reads have been deposited with the NCBI (GenBank accession No. SRP153785).

## Sequence data processing and de novo assembly

The raw reads generated by Illumina sequencing were subjected to a screening process where the adapter sequences and low-quality reads were trimmed from the raw reads using Trim Galore v0.5.0 (http://www.bioinformatics.babraham.ac.uk/projects/trim_galore/) software. FastQC v0.11.8 (http://www.bioinformatics.babraham.ac.uk/projects/fastqc/) was used to check the quality of the pretreated data. Trinity v2.8.4 (https://github.com/trinityrnaseq/trinityrnaseq/releases/tag/Trinity-v2.8.4) was used to perform de novo assembly with the default parameters (*Grabherr et al., 2011*). The clean high quality data were spliced to obtain the reference sequences (transcripts) for subsequent analyses.

## Gene annotation

Multiple public databases were used for homology annotation, including non-redundant nucleotide and protein sequences (NT/NR) (http://www.ncbi.nlm.nih.gov), Swiss-Prot

(http://www.expasy.ch/sprot), euKaryotic Orthologous Groups (KOG) (https://ftp.ncbi.nih.gov/pub/COG/KOG/) based on BLASTX (NCBI blast 2.2.28) with an *e*-value of 1*e*-5 (*Altschul et al., 1990*), Gene Ontology (GO) (http://www.geneontology.org/) based on Blast2GO v2.5 with an *e*-value of 1*e*-6, and Kyoto encyclopedia of genes and genomes (KEGG; http://www.genome.jp/kegg) based on KAAS with an *e*-value of 1*e*-10 (*Kanehisa et al., 2008*). InterProScan v5.31-70.0 (http://www.ebi.ac.uk/interpro/download.html) was also used to predict the functional domains, signal peptides, and other protein characteristics by BLASTing against the Conserved Domain Database Interpro under the default parameters (*Zdobnov & Apweiler, 2001*). GO terms were assigned to three categories (biological process, molecular function, and cellular component) (*Conesa et al., 2005*). The KEGG Automatic Annotation Server was employed for KEGG annotation (*Moriya et al., 2007*).

## DEG analysis

Clean reads were aligned using Bowtie 2 v2.3.4.3 (http://bowtie-bio.sourceforge.net/bowtie2/index.shtml) (*Langmead, 2010*). Transcript abundance was estimated using RSEM v1.3.1 (http://deweylab.github.io/RSEM/) in the Trinity package (*Li & Dewey, 2011*). The expression profiles were detected for the transcripts using the gene expression method as the fragments per kilobase of transcript per million mapped reads (*Trapnell et al., 2010*). For samples with three biological replicates, the differential expression of unigenes between males and females was analyzed using the R (v3.4.2) package DESeq (*Anders & Huber, 2010*). Based on the negative binomial distribution model, DESeq provides statistical routines for determining differential expression in digital gene expression data (*Hu et al., 2018*). The statistical *P*-values were adjusted using Benjamini and Hochberg's approach for controlling the false discovery rate (*Benjamini & Hochberg, 1995*). Unigenes with a corrected *P*-value < 0.05 and $\log_2$FoldChange $\geq 1$ or $\log_2$FoldChange $\leq -1$ were considered to be DEGs. Significant specific-expressed genes (SEGs) were identified from DEGs. Visualizations of the analyses, including the heatmap, volcano, GO, and KEGG enrichment results, were performed in the R package with ggplot2.

## Validation of transcriptome data by real-time qRT-PCR

To verify the accuracy of the transcriptome data, three DEGs and nine SEGs were selected randomly for real-time qRT-PCR. All of the reactions were performed using three technical replicates and three biological replicates to validate the reliability of the results. A Real-time Detection System (C1000 Thermal Cycler; Bio-Rad, Hercules, CA, USA) was employed to perform real-time qRT-PCR. The PCR reaction system comprised the following: 10 μL SuperReal PreMix Plus (Tiangen, Beijing, China), 0.5 μL of each primer (10 μM in total), one μL of template cDNA, and eight μL of RNase-free ddH$_2$O. The expression profile was detected in triplicate wells under the following protocol: 95 °C for 15 min, 40 cycles for 10 s at 95 °C and 20 s at 58 °C, and 30 s at 72 °C, before finally ramping from 65 to 95 °C at 0.5 °C per 5 s to generate a melting curve. The turtle *GADPH* gene was selected as the internal control gene. The relative expression level of each

**Table 1 Statistical analysis of the assembly quality for *M. reevesii* transcriptomes.**

| Sample | Male 1 | Male 2 | Male 3 | Female 1 | Female 2 | Female 3 | Total |
|---|---|---|---|---|---|---|---|
| Raw reads | 81,369,826 | 71,156,674 | 69,466,956 | 50,354,874 | 124,062,604 | 112,374,346 | 508,785,280 |
| Raw bases, Mbp | 12,205 | 10,673 | 10,420 | 7,553 | 18,609 | 16,856 | 76,316 |
| Trim reads | 79,459,524 | 69,492,528 | 67,807,520 | 49,205,254 | 121,068,366 | 109,130,658 | 496,163,850 |
| Trim bases, Mbp | 11,591 | 10,124 | 9,889 | 7,161 | 17,668 | 15,969 | 72,402 |
| Average length | 145.88 | 145.69 | 145.84 | 145.54 | 145.94 | 146.33 | 145.87 |
| Trim reads, % | 97.65 | 97.66 | 97.61 | 97.72 | 97.59 | 97.11 | 97.57 |
| Trim bases, % | 94.97 | 94.86 | 94.91 | 94.81 | 94.94 | 94.74 | 94.87 |

**Note:**
Testis: Male 1, Male 2, Male 3; Ovary: Female 1, Female 2, Female 3.

contig in different samples was calculated using the $2^{-\Delta\Delta CT}$ method (*Livak & Schmittgen, 2001*). The primers employed are listed in the supplemental materials (Table S1).

# RESULTS

## Sequencing and de novo assembly

Six libraries (3♂, 3♀) were established and sequenced (Table 1). In total, 508,785,280 paired-end raw reads with lengths over 100 bp were generated. We obtained 496,163,850 (97.57%) high-quality reads with an average length of 145.87 bp. The de novo assembly results yielded 557,690 isoforms, which clustered into 437,767 unigenes with an N50 of 631 bp. The GC content of the entire final assembly was 53.46%.

## Unigene functional annotation

Among the total of 437,767 unigenes, 188,633 unigenes were annotated with the seven databases. The results are shown in Table S2. Among the databases, 20.08% of the unigenes were aligned to the Nr protein database with an *e*-value threshold of *e*-5 (Fig. S1).

In total, 17,361 GO annotations were assigned to three terms: biological processes (12,137 annotations), molecular functions (1,379 annotations), and cellular component (3,845 annotations). Among the biological processes, cellular, metabolic, organic substance metabolic, and primary metabolic processes were the most highly represented. The majority of the proteins assigned to molecular functions were associated with binding and catalytic activity. Among the cellular components, cell and intracellular proteins were the most highly represented (Fig. S2).

KEGG pathway analysis annotated 12,456 transcripts into 320 pathways, where 24.6% of all the annotated transcripts were related to metabolism, and 1,922 transcripts (15.4%) were involved with signaling molecules and interaction. The cytokine–cytokine receptor interaction pathway (ko04060; 804 sequences) was the most highly represented.

## Analysis of DEGs

In total, 36,221 transcripts could be detected in the combined sex transcriptomes. After strict filtration, 756 DEGs were upregulated in the testis and 838 DEGs in the ovary (Fig. 2). The DEGs are listed in Table S3. A heatmap illustrating the hierarchical
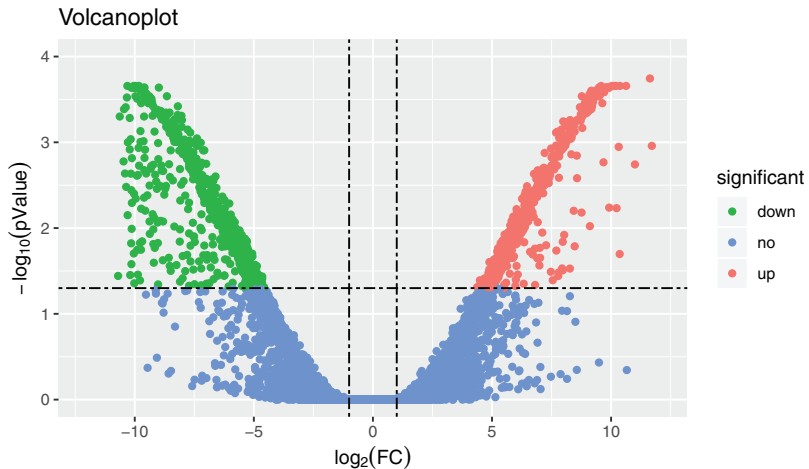

**Figure 2** **Volcanoplot where *x*-axis represents the level of differential expression and the *y*-axis shows the significant differences in expression as negative log values.** The horizontal line is the threshold of corrected *P*-value < 0.05. Downregulated genes in the ovaries are indicated by green dots, upregulated genes in the ovaries are indicated by red dots, and other genes are indicated by blue dots.

clustering of the DEGs was generated to visualize the overall gene expression pattern (Fig. 3). Among the 1,594 DEGs, we found nine significant SEGs, as shown in Table 2.

## Enrichment analysis of DEGs

The 1,175 GO terms were annotated as biological processes (859; 73.1%), cellular component (109; 9.3%), and molecular function (207; 17.6%). DEGs were assigned to 39 processes by KEGG enrichment analysis, including Cellular Processes (310 DEGs, 22.27%), Metabolism (88 DEGs, 22.27%), Genetic Information Processing (216 DEGs, 22.27%), Environmental Information Processing (352 DEGs, 22.27%), Human Diseases (662 DEGs, 22.27%), and Organismal Systems (288 DEGs, 22.27%). We used the same strategies (*Du et al., 2017*) used to identify genes and pathways related to gonadal development and the gametogenesis system, and 20 significantly enriched GO terms were detected in the ovary and testis. These enriched sex-related terms included male gonad development, tube development in testis, embryo development, iron ion binding, and regulation of signal transduction in ovary. The 20 GO terms are shown in Fig. 4. In addition, 10 KEGG terms were significantly enriched in the ovary and testis, that is, the Hedgehog signaling pathway, pathways in cancer, TGF-beta signaling pathway, prostate cancer in testis, metabolic pathways, Hippo signaling pathway, ovarian steroidogenesis, steroid hormone biosynthesis, Wnt signaling pathway, and oocyte meiosis in ovary. The 10 KEGG terms are shown in Fig. 5.

## Validation of transcriptome data by real-time qRT-PCR

A total of 12 genes comprising six ovary SEGs (*FGF9*, *FOXL2*, *GDF9*, *WNT4*, *CYP19A1*, and *SAMD9*), three testis SEGs (*SOX9*, *AMH*, and *AR*), and three DEGS (*ARHGEF9*, *CKAP5*, and *PTH2R*) were selected for validation using qRT-PCR in order to confirm

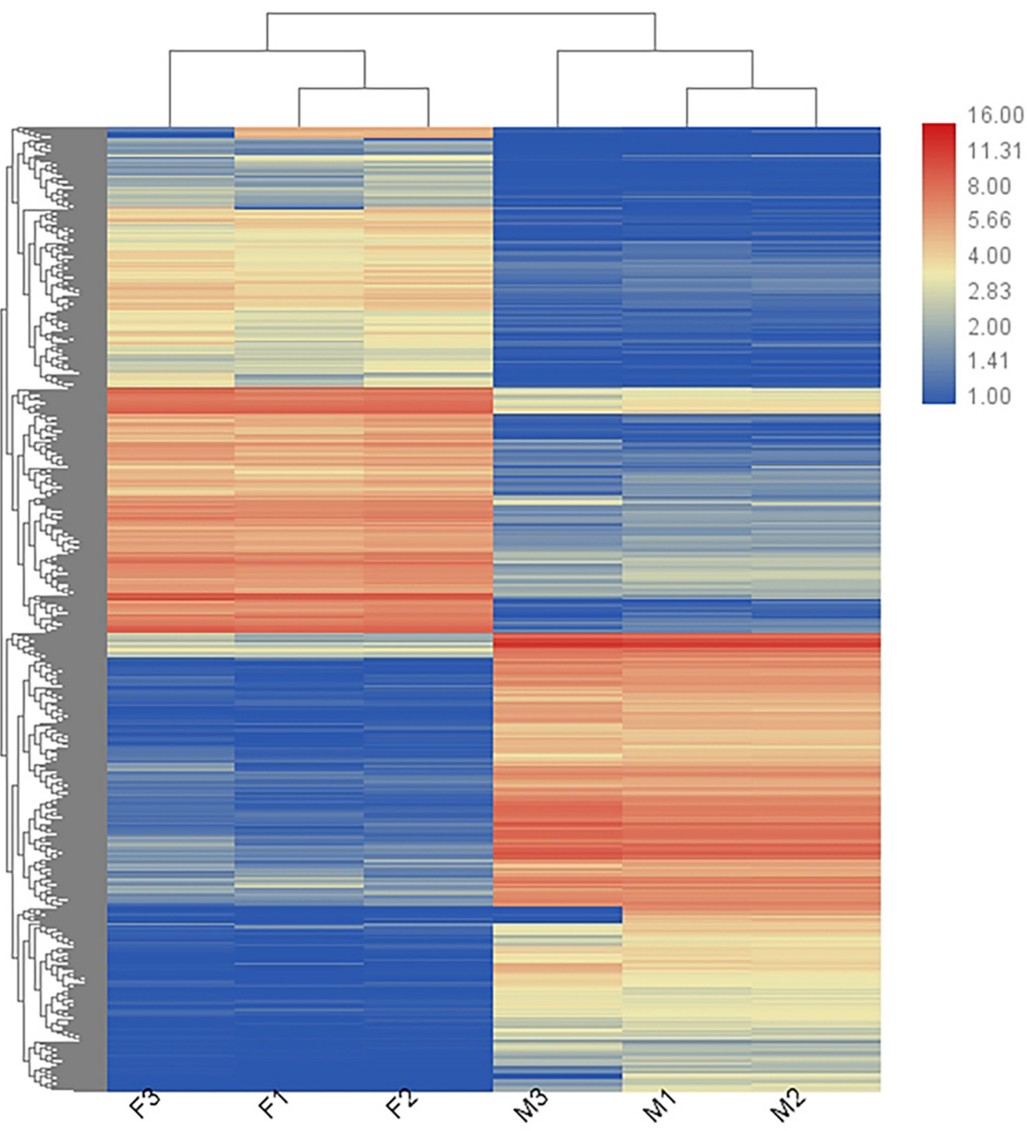

**Figure 3 Heatmap showing the DEGs in males (M1, M2, and M3) and females (F1, F2, and F3).** Blue represents weakly expressed genes and red represents highly expressed genes.

the reliability and accuracy of the RNA-Seq method (Fig. 6). The results suggested that all of the genes had similar expression patterns according to both RNA-Seq and qRT-PCR, which also confirmed that RNA-Seq accurately quantified the expression of genes in the ovaries and testes.

## DISCUSSION

In this study, we conducted the first transcriptomic gene expression analysis of adult gonads in a turtle species with TSD. In the gonads, we identify testis- and ovary-specific genes, and defined the downstream gene expression cascades for both the male and female pathways (Fig. 7).

**Table 2 Annotation of the significant SEGs.**

| Gene name | Ensembl ID | Description | Testis_ average | Ovary_ average | log$_2$FoldChange | Q value |
|---|---|---|---|---|---|---|
| CYP19A1 | TR102156\|c1_g2_i2 | Aromatase | 3.94 | 492.55 | 6.64 | 4.58E-03 |
| FOXL2 | TR138024\|c0_g1_i2 | Forkhead box protein L2 | 0.52 | 660.25 | 8.77 | 2.86E-04 |
| AMH | TR109517\|c0_g1_i1 | Anti-Müllerian hormone | 1,526.67 | 51.75 | −4.86 | 4.44E-02 |
| GDF9 | TR148262\|c0_g3_i1 | Growth/differentiation factor 9 | 21.29 | 11,568.91 | 9.02 | 4.45E-04 |
| FGF9 | TR108256\|c0_g1_i4 | Fibroblast growth factor 9 | 9.79 | 442.85 | 5.36 | 2.25E-02 |
| SMAD9 | TR137938\|c0_g1_i2 | Mothers against decapentaplegic homolog 9 | 10.3 | 339.08 | 4.91 | 3.53E-02 |
| WNT4 | TR167755\|c2_g1_i1 | Wingless-related MMTV integration site 4 | 105.49 | 7,614.18 | 6.16 | 7.14E-03 |
| SOX9 | TR144387\|c0_g2_i1 | Transcription factor SOX-9 | 1,637.35 | 54.36 | −4.89 | 4.05E-02 |
| AR | TR166731\|c0_g1_i3 | Androgen receptor | 32.95 | 0.26 | −4.75 | 4.37E-02 |

Note:
Testis_ average: the average gene expression values of three testis biological replicates; Ovary_ average: the average gene expression values of three testis biological replicates; log$_2$FoldChange: the relative expression level of genes in ovary compared to that in testis (log2 transformation); Q value: corrected *P*-value.

## Existence of antagonistic signals and pathways related to gonadal maintenance and gametogenesis in *M. reevesii*

In the sexual maturity period, steroid hormones such as estrogen and androgen are necessary for gonadal maintenance based on the sexually dimorphic expression of *CYP19A1* and *AR*. *CYP19A1*, which is a gene encoding an enzyme that catalyzes conversion from androgens to estrogens (*Piferrer, 2013*; *Matsumoto et al., 2013*). *CYP19A1* and other important genes such as *SOX9* and *FOXL2*, which are related to sex determination and differences, exhibited sex-dimorphic expression in the early thermosensitive period (TSP) (*Tang et al., 2017*), and they still played core roles in the adult gonads in *M. reevesii*. We also identified conserved genes with known functions in sexual differentiation comprising *AMH*, *GDF9*, and *WNT4*. These genes also had obvious sexually dimorphic expression patterns. The male-specific expression of *SOX9* mRNA during fetal and adult life indicates that it has a fundamental role in testis development in the turtle (*Kent et al., 1996*; *She & Yang, 2017*). In turtles, *AMH* is regulated by *SOX9* and this pattern is consistent with *SOX9* upregulating *AMH* in the mammalian testis (*Bieser & Wibbels, 2014*). Typical ovarian-specific typical markers (*FOXL2* and *CYP19A1*) and testicular-specific marker *SOX9* are often expressed in a mutually exclusive manner in the gonads (*Matsumoto & Crews, 2012*). *FOXL2* suppresses *SOX9* transcription by cooperatively binding with estrogen receptors in the regulatory region of *SOX9* (*Uhlenhaut et al., 2009*). *GDF9* can regulate a wide range of activities in the granulosa and theca cells, including the secretion of steroid hormones (*Elvin, Yan & Matzuk, 2000*), and it significantly suppresses the expression of *AMH* as well as being co-expressed with *CYP19A1* during the sexual maturity period (*Wang et al., 2017*). *WNT4* plays a key role in the female sexual development pathway by controlling steroidogenesis in the gonads and possibly supporting oocyte development (*Vainio et al., 1999*), as well as repressing typical male-specific steroidogenesis (*Bernard & Harley, 2007*).

*DMRT1* is a candidate master male sex-determining gene in *T. scripta* with TSD during early gonad development (*Ge et al., 2017*). However, in the adult turtle gonads, *DMRT1* was detected at low levels and it did not exhibit transcriptional sexual dimorphism
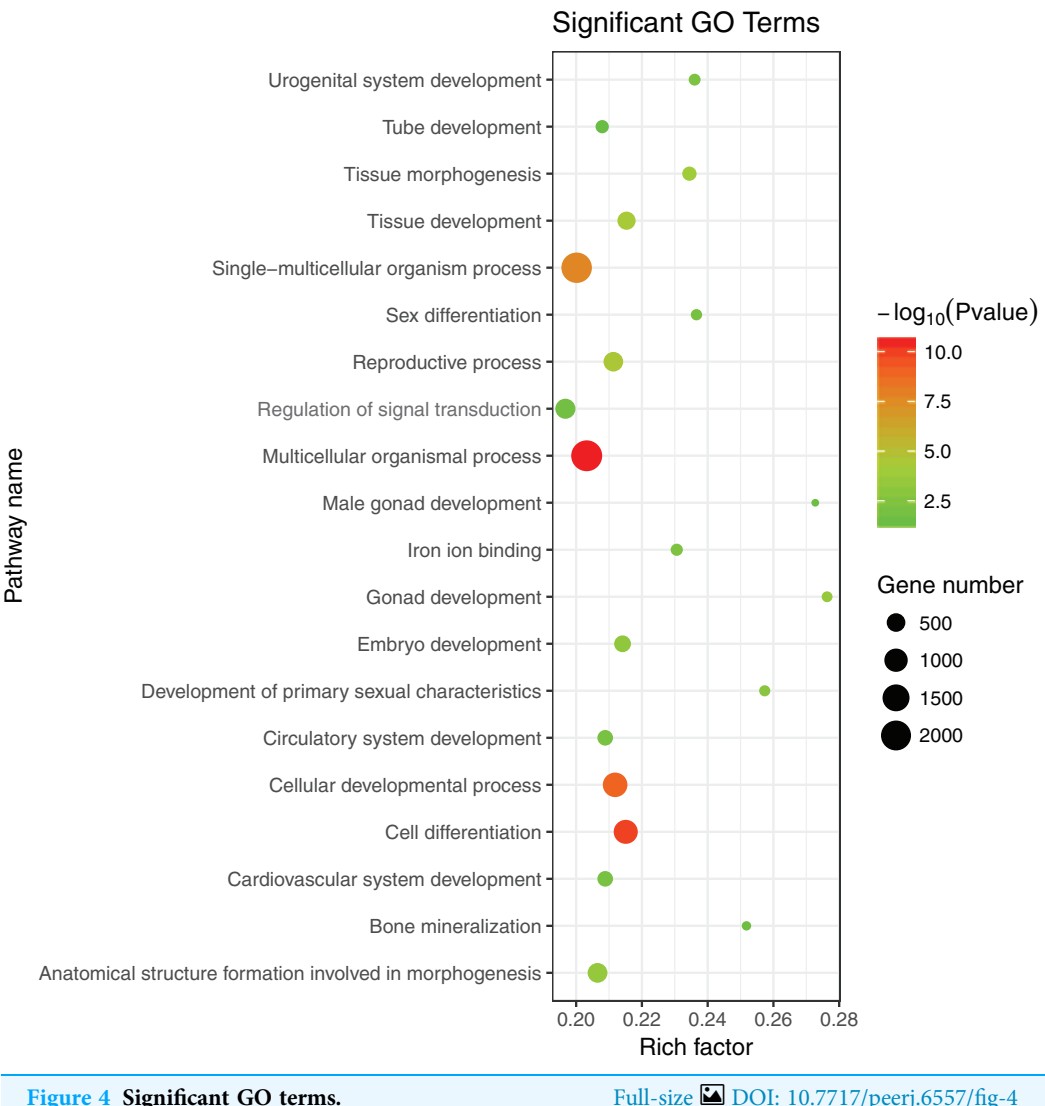

**Figure 4 Significant GO terms.**

between the testis and ovary. Our results suggest that *DMRT1* does no play an important role in the adult differentiated gonads. *Czerwinski et al. (2016)* identified six male-producing temperature-enriched genes and four female-producing temperature-enriched genes with differences in expression prior to *DMRT1* and *CYP19A1* in the early TSP. The detection of these 10 genes suggests that a gonad-specific regulatory mechanism exists but it is not clear if it is causal. In our study, *NOV*, *KDM6B*, *RBM20*, *PCSK6*, *AVIL*, and *TWIST1* were detected at low levels and their expression levels did not differ between the adult testis and ovary. *HSPB6* was not detected. *VWA2* and *FDXR* were upregulated in the ovary, but their expression levels were low in the testis, and only *FANK1* was strongly upregulated in the adult testis. Our results suggest that the potential gonad-specific regulatory mechanism might differ between the embryo and adult.

The Hippo signaling pathway and Wnt signaling pathway were significantly enriched in the ovary, whereas the TGF-beta signaling pathway and Hedgehog signaling pathway
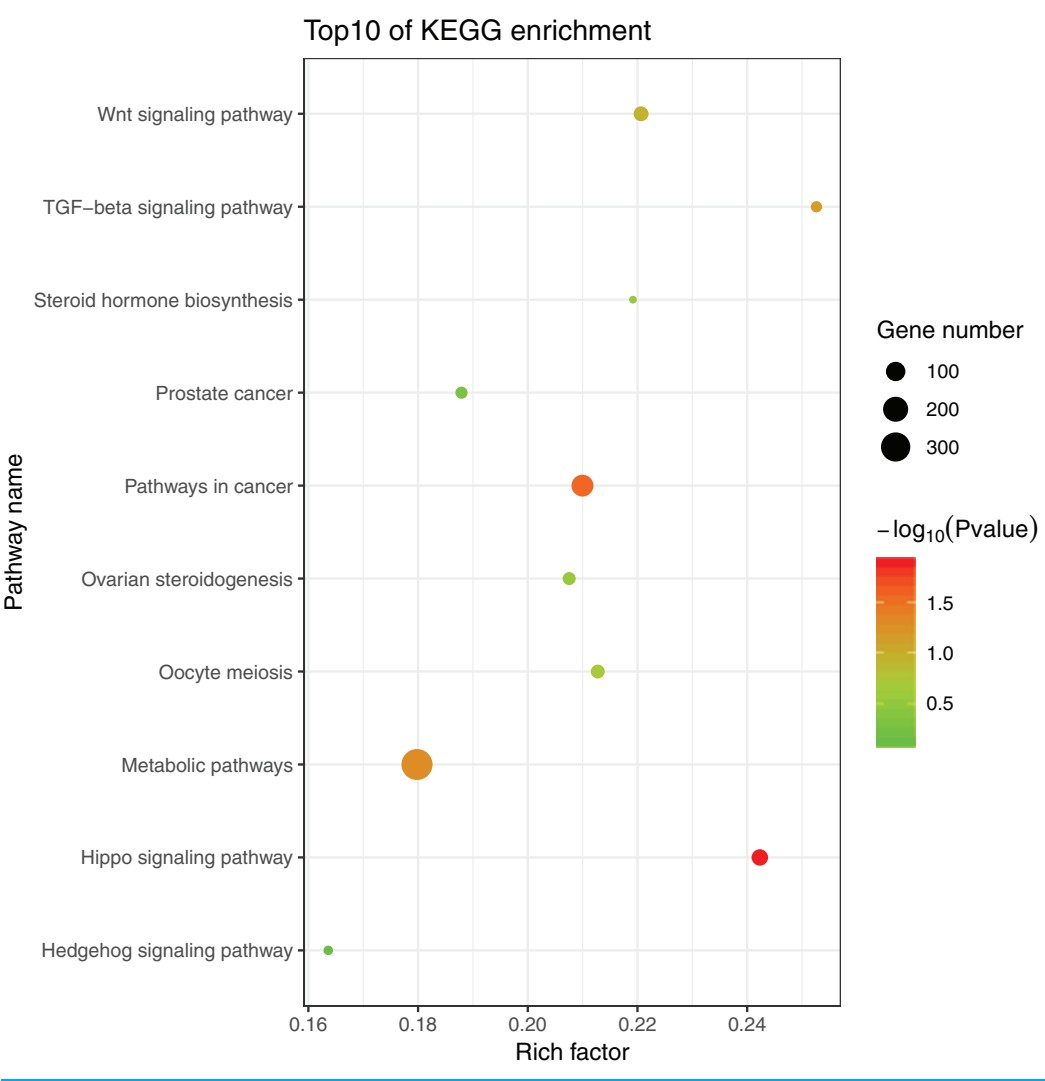

Figure 5 **Significant KEGG terms.**

were significantly enriched in the testis (Fig. 7). The Hippo signaling pathway has critical roles in controlling cell proliferation, self-renewal, differentiation, and apoptosis in most tissues and organs in diverse species (*Lyu et al., 2016*) and it represents a molecular target for the regulation of mouse ovarian functional remodeling that could be used to regulate the proliferation and differentiation of ovarian function (*Ye et al., 2017*). Our analysis of *M. reevesii* suggested that Wnt signaling appears to function downstream of estrogen and the ovary-promoting effects of the Wnt signaling pathway may be functionally conserved in mammals and reptiles. Activation of the Wnt pathway antagonizes nuclear *SOX9* expression (*Mork & Capel, 2013*). The TGF-beta signaling pathway plays an important role in the differentiation of male germ cells in non-mammalian vertebrates (*Zhang et al., 2016*), for example, *AMH* is a growth factor in the TGF-beta family with a central role in testis formation. The Hedgehog signaling pathway has crucial roles in the development of diverse tissue and organ systems in the embryo, and in the regulation of adult tissues (*Migone et al., 2017*).
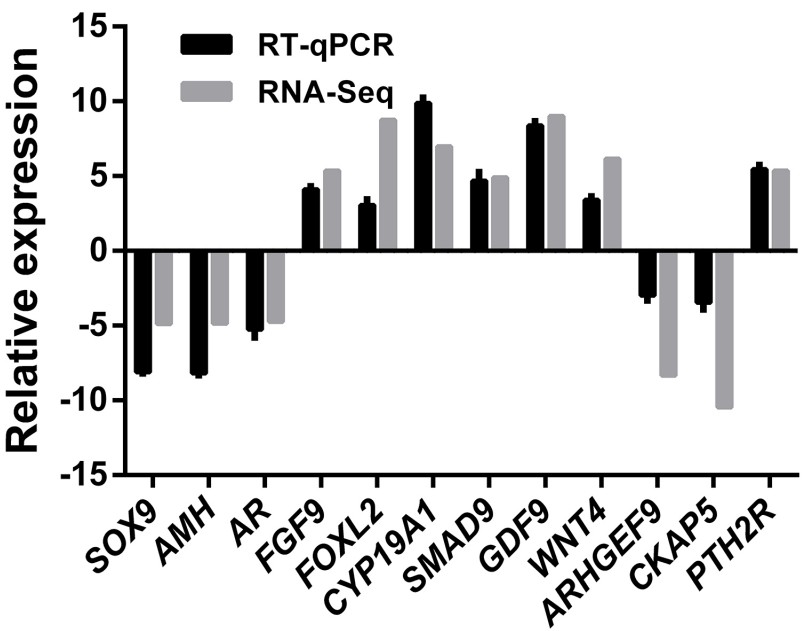

**Figure 6 RNA-Seq and qRT-PCR validation results.** Expression comparisons of selected genes detected by RNA-Seq and qRT-PCR between ovary and testis. The *y*-axis show log2 (fold differences) determined by RNA-Seq and qRT-PCR. The experiments were repeated three times and provided consistent results. The mean values and error bars were obtained from three biological and three technical replicates.

**Figure 7 Putative genes and pathways involved in testis and ovary functional maintenance.**

## Candidate gene *FANK1* might be involved with the regulation of testis spermatogenesis

*FANK1* is exclusively expressed in the testis from the meiosis phase to the haploid phase of spermatogenesis in mice, and it may have a crucial functional role in spermatogenesis as a transcription factor (*Dong et al., 2014*). Indeed, reduced sperm numbers and increases in apoptotic spermatocytes were observed in *FANK1* knockdown mice. In *M. reevesii*, *FANK1* was strongly upregulated in the adult testis. Thus, it is possible that *FANK1*

play a pivotal role in spermatogenesis as a transcription factor in the adult gonads. Future studies should investigate the function of *FANK1* during the regulation of testis maintenance.

## CONCLUSIONS

According to our results, we propose a model where the genetic components related to steroid hormones comprise networks with multiple feedback loops. The networks are antagonistic in males and females, where differences in gene expression accumulate and converge to allow the antagonistic regulation of steroid hormones to maintain the appropriate balance in males and females. *FANK1* could be involved with the regulation of testis spermatogenesis in the adult turtle gonads.

## ACKNOWLEDGEMENTS

The authors are grateful for the comments provided by the reviewers and editors regarding the manuscript.

### Funding

This research was supported by the National Natural Science Foundation of China (NSFC, No. 31372198), the Research Fund of the Key Laboratory of Biotic Environment and Ecological Safety of Anhui province, the Natural Science Research Project in Colleges and Universities of Anhui province (No. KJ2017A254). The funders had no role in study design, data collection and analysis, decision to publish, or preparation of the manuscript.

### Grant Disclosures

The following grant information was disclosed by the authors:
National Natural Science Foundation of China: 31372198.
Research Fund of the Key Laboratory of Biotic Environment and Ecological Safety of Anhui province.
Natural Science Research Project in Colleges and Universities of Anhui province: KJ2017A254.

### Competing Interests

The authors declare that they have no competing interests.

### Author Contributions

- Lei Xiong conceived and designed the experiments, performed the experiments, analyzed the data, contributed reagents/materials/analysis tools, prepared figures and/or tables, authored or reviewed drafts of the paper, approved the final draft.
- Jinxiu Dong prepared figures and/or tables, approved the final draft.
- Hui Jiang prepared figures and/or tables, approved the final draft.
- Jiawei Zan contributed reagents/materials/analysis tools, approved the final draft.
- Jiucui Tong performed the experiments, approved the final draft.

- Jianjun Liu analyzed the data, contributed reagents/materials/analysis tools, approved the final draft.
- Meng Wang contributed reagents/materials/analysis tools, approved the final draft.
- Liuwang Nie conceived and designed the experiments, authored or reviewed drafts of the paper, approved the final draft.

## Animal Ethics

The following information was supplied relating to ethical approvals (i.e., approving body and any reference numbers):

Procedures involving animals and their care were approved by the Animal Care and Use Committee of Anhui Normal University (#20170612).

## Data Availability

The Illumina reads are available in the Sequence Read Archive (SRA) database at NCBI under study accession number SRP153785.

## Supplemental Information

Supplemental information for this article can be found online at http://dx.doi.org/10.7717/peerj.6557#supplemental-information.

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
