# Peer review of "Transcriptome sequencing and comparative analysis of adult ovary and testis identify potential gonadal maintenance-related genes in Mauremys reevesii with temperature-dependent sex determination"

_PeerJ, doi:10.7717/peerj.6557_

## Round 0.1 · original submission · Major Revisions

The premise of this paper is interesting and RNA seq data are collected and analyzed appropriately. However, I am in agreement with reviewer 1 in that the introduction (and some of the discussion) is not aligned with the question and supporting data. Temperature sensitive sex determination occurs during embryonic and early larval development, and the authors do not make a strong connection between the established mechanisms occurring during early life and the genomic data they have generated and analyzed in adults. Because the RNA seq work is of interest and is novel for this species, I believe the manuscript to have some merit. That being said, it is crucial that the authors reframe the paper as to focus on the importance of gonadal maintenance in organisms with TSD, with more up-to-date references. Because embryos were not analyzed, there should be very little emphasis on mechanisms specific to embryonic and early larval sex determination. For example, one of the main supporting papers cited in the introduction is from 2009, nearly 10 years ago. The discussion thoroughly describes DEGs and their relevance to other studies, but not in the context of TSD. Readers really need to be convinced of the importance of gonadal maintenance and and its relevance to TSD, with stronger connections to current TSD literature across life stages with an emphasis on events occurring after maturity.

Reviewer 1 ·

Basic reporting

The article is difficult to read and the motivations for the study is unclear. The introduction focuses on previous studies into the mechanism of temperature dependent sex (TSD) in reptiles. Sex determination & TSD occurs in the gonads of the developing embryo and, therefore, previous studies have investigated gene expression in the embryonic gonad (eg. Ge et al 2018, Czerwinski et al 2017 & Yatsu et al 2017). In the present manuscript, the authors measure gene expression in adult gonads of a turtle species. This has little or no relevance to the mechanism of TSD, since TSD occurs in the embryo. The authors instead claim to be investigating mechanisms of 'gonadal maintenance and gametogenesis'. If this is the case, the authors should provide background on this subject in the introduction, rather than discussing TSD, since their data has little relevance to TSD. At present, I am unsure what the authors mean by 'gonadal maintenance and gametogenesis' and why they chose to investigate this subject.

Experimental design

The experimental design is suitable for a comparison of gene expression between adult testis and ovary in this turtle species. However, the experimental design is not suitable fr studying TSD, since this occurs in the embryo (see above comments).

Validity of the findings

The authors have generated a data resource that may be useful for future studies in this species. However, similar data has been previously generated for other turtle species and other reptile species, so the novelty of the dataset is small. Likewise, their analysis has not yielded any original findings - they have simply characterised a number of genes/pathways with previously known roles in sex/gonad identity. In my view, more detailed investigation of a more specific hypothesis is required to be suitable for publication.

Reviewer 2 ·

Basic reporting

This manuscript offers insights into molecular mechanisms for sex determination in an organism with TSD. However, the background and discussion pertaining to current literature regarding molecular mechanisms of TSD should be expanded on. Specifically, there is very little mention of the importance of the aromatase genes (CYP19a and CYP19b) in regulation of TSD. This mechanism is well studied in reptiles as well as other species, and should be included in the introduction.

The introduction or discussion would also benefit from adding more information on the model organism (i.e. common name, optimal temperature for 50:50 sex ratios, and temperatures at which male/female sex is determined).

There are a few sentences (lines 59-62; lines 74-79) that could be revised or restructured for clarification as they are difficult to understand as written.

Experimental design

The authors present a robust study design, validating findings with multiple methods of quantifying gene expression. However, there are a few key points that need clarification.

More information is need on the rearing conditions of the animals used in this study. Is it possible that their sexual determination could have been influenced by temperature at early life stages? Were the animals sampled independently?

Was expression of genes compared between males and females as suggested by Table 2? More information is needed in the methods on how these comparisons were made.

Validity of the findings

More information is needed pertaining to statistical tests and comparisons made.
Line 136: Specifically what types of tests were used to make comparisons? Were differences detected between sexes or relative to the reference gene? A sample size should also be included here. Were comparisons made using any other data sets in this study?

Calculated efficiency values for primers should be added to Table S1.

Lines 151-152: Was this based on a manufacturer protocol? Why not use the standard 40 cycles?

Additional comments

Citations are needed on a couple sentences in the manuscript:
Lines 152-153: Citation is needed for the paper describing this method
Lines 213-215: Citation is need for this information

Figure 3: It seems that F2 is presented in a continuous color gradient and all other samples are shown in a discrete color gradient? If this is meant to illustrate a particular finding, more information should be given in the figure caption.

Table 2: More information is needed on the tests (sample size, which groups were compared) used to obtain the Q-value in the table caption.

This paper presents interesting insights into the mechanisms of sexual determination. I think this paper would be strengthened by expanding upon the discussion of current literature pertaining to mechanisms of sexual determination in organisms with TSD and including more information on the animals used in this study.

---

## Round 0.2 · accepted · Accept

Thank you for thoroughly addressing both reviews.

# Reviewer 1 ·

Basic reporting

The revisions have greatly improved the readability the article, and the motivations of the study (to investigate gene expression patterns involved in gonad maintenance) are more clearly articulated.

Experimental design

My reservations about the experimental design have been resolved by shifting the focus of the article from TSD to gonadal maintenance, since adult tissues were studied. The experimental design is now suitable, in my opinion, for the question being addressed.

Validity of the findings

As above, I did not have reservations about the experiment or its findings, but about their interpretation. By changing the focus from TSD to gonad maintenance, the authors have resolved my concerns.

Additional comments

I think the article has been improved by the revisions and would be happy to see it published.

Reviewer 2 ·

Basic reporting

This version of the manuscript was improved for clarity throughout.

Experimental design

While most of my concerns were addressed, I still think the methods are lacking on information about the animals (i.e. age, reproductive status).

Validity of the findings

All of my previous concerns have been addressed.

Additional comments

This version of the manuscript is much improved, however more information is needed on the study design particularly about the animals sampled. As is, the study design would be very difficult to reproduce.